# Competitive Effects of Oxidation and Quantum Confinement on Modulation of the Photophysical Properties of Metallic-Phase Tungsten Dichalcogenide Quantum Dots

**DOI:** 10.3390/nano13142075

**Published:** 2023-07-15

**Authors:** Bo-Hyun Kim, Jun Yong Yang, Kwang Hyun Park, DongJu Lee, Sung Ho Song

**Affiliations:** 1Division of Advanced Materials Engineering, Center for Advanced Powder Materials and Parts, Kongju National University, Cheonan 32588, Republic of Korea; bohkim@kongju.ac.kr (B.-H.K.); yajy2306@naver.com (J.Y.Y.); recite14@gmail.com (K.H.P.); 2Department of Advanced Materials Engineering, Chungbuk National University, Chungdae-ro 1, Seowon-gu, Cheongju 34057, Republic of Korea

**Keywords:** quantum dots, transition metal dichalcogenide, quantum confinement, photoluminescence, optical property

## Abstract

Metallic-phase transition metal dichalcogenide quantum dots (TMDs-*m*QDs) have been reported in recent years. However, a dominant mechanism for modulating their intrinsic exciton behaviors has not been determined yet as their size is close to the Bohr radius. Herein, we demonstrate that the oxidation effect prevails over quantum confinement on metallic-phase tungsten dichalcogenide QDs (WX_2_-*m*QDs; X = S, Se) when the QD size becomes larger than the exciton Bohr radius. WX_2_-*m*QDs with a diameter of ~12 nm show an obvious change in their photophysical properties when the pH of the solution changes from 2 to 11 compared to changing the size from ~3 nm. Meanwhile, we found that quantum confinement is the dominant function for the optical spectroscopic results in the WX_2_-*m*QDs with a size of ~3 nm. This is because the oxidation of the larger WX_2_-*m*QDs induces sub-energy states, thus enabling excitons to migrate into the lower defect energy states, whereas in WX_2_-*m*QDs with a size comparable to the exciton Bohr radius, protonation enhances the quantum confinement.

## 1. Introduction

Two-dimensional layered transition metal dichalcogenides (TMDs) are known to have two different phases, semiconducting and metallic (1T/1T’), which induce miscellaneous changes in their electronic and optical properties [1,2,3]. Unlike semiconducting TMDs, metallic-phase (1T/1T’) TMDs and their QDs show a feature of high charge transfer efficiency for photocatalytic and electrocatalytic abilities [4,5,6]. Recently, two sizes of metallic-phase TMDs-QDs (TMDs-*m*QDs) were demonstrated to have the photophysical properties of different charge excitation and decay pathways, which was explained by defect states and valance band splitting [3]. The variation in the optical energy structure in QDs is generally ascribed to quantum confinement and edge functionality [7,8,9]. However, the dominant mechanism that intrinsically controls exciton behaviors as the size and environmental conditions change remains elusive, although this issue is important for utilizing the photophysical properties in practical applications.

Modulating the optical properties of semiconducting TMDs-QDs has been conducted by using size control [3,4,10,11,12], environmental solvents [13,14,15], surface/edge functionalization [16], defect control [17], heterostructures [18], and structural phase control [5]. These modulation techniques have found applications across various fields [19,20]. Their photophysical properties are known to be affected by quantum confinement, surface defects, pH-sensitive edge-functional groups, and structural phases. The semiconducting WS_2_-QDs with a size of ~2 nm, where a 2–4 nm size is comparable to the exciton Bohr exciton radius [21,22], showed strong emission which was ascribed to quantum confinement [23]. From the excitation-wavelength-dependent PL of WS2-QDs with diameters of 1.5–3.5 nm, the localized states were suggested to be the origin of PL [4]. The high quantum yield (~5.5%) in the dimension- and phase-modulated WS_2_-QDs with a size of ~2.7 nm was attributed to the suppression of vibrational modes in WS_2_-QDs [24]. Despite these experimental results showing enhanced PL properties in the TMDs-QDs with a size comparable to the exciton Bohr radius, it is still vague whether quantum confinement is a dominant mechanism in TMDs-QDs with a size larger than the exciton Bohr radius and with an edge structure sensitively varied by pH change and/or functional groups. It was reported that MoSe_2_-QDs and MoS_2_-QDs have a pH-dependent PL [12,25], but the pH-independent PL property of WSe_2_-QDs was also demonstrated [26]. These contrasting results are controversial and have not been explained yet. As the 1T/1T’ metallic-phase structure of TMDs is thermodynamically metastable compared to the stable 2H semiconducting phase structure [6], the optical properties of the TMDs-*m*QDs remain to be investigated.

In this work, we investigate the difference in the photophysical properties of metallic phase-tungsten dichalcongens QDs (WX_2_-*m*QDs) of two sizes (diameters of ~3 nm and ~12 nm) dispersed in water with pH 2 and 11. We measure the change in the optical properties of WX_2_-*m*QDs (X = S and Se) depending on the changes in size and the pH of the solution using UV-Vis absorbance, PL, excitation-energy-dependent PL (PLE), and time-resolved PL (TRPL). Based on our comparison between the results, we discuss the primitive function on the photoelectric energy structure and exciton dynamics for WX_2_-*m*QDs with sizes varying from ~3 nm to ~12 nm.

## 2. Materials and Methods

Potassium sodium tartrate was selected to synthesize WX_2_-*m*QDs at low temperature to minimize damage [3,27,28]. To fabricate the WX_2_-*m*QDs, we followed the previous process for GQDs and TMD-QDs in Refs. [5,22], and the details can be found there. Briefly, the initial steps started with the mixing and grinding of potassium sodium tartrate (200 mg) with WS_2_ (WSe_2_) (20 mg) purchased from Sigma Aldrich in Seoul, Republic of Korea. Then, the ground homogeneous mixtures were reacted in an autoclave vessel at 250 °C for 12 h, and then instantly exfoliated in water with sonication. The sizes of the WX_2_-*m*QDs were firstly controlled by AAO (20 nm, Whatman, Merch KGaA, Republic of Korea) filtration, and then dialyzed using dialysis tubing (10,000 and 8000 NMWL, Amicon Ultra-15, Merck KGaA, Seoul, Republic of Korea) to separate them into two sizes, which simultaneously removed the remaining salts. Finally, the WX_2_-*m*QDs were obtained as a dispersed solution in water. After drying, the WX_2_-*m*QDs were re-dispersed with an appropriately low concentration (<0.1 mg/mL) in 10 mL of water with pH 2 and 11.

The morphologies of the *m*QDs were analyzed using an AFM (X2-70, Park Systems Corp, Suwon, Republic of Korea) in tapping mode under ambient conditions at the Smart Nature Research Centre. XPS (Sigma Probe, AlKα), UV/Vis spectra (UV-3600, Shimadzu spectrometer, Smart Nature Research Centre, Cheonan, Republic of Korea), fluorescence spectra (Perkin-Elmer LS 55 luminescence spectrometer, PerkinElmer, MA, USA), and transmission electron microscopy (TEM, Titan cubed G2 60-300, FEI, OR, USA) analyses were conducted. PL measurements were carried out using a 325 nm He-Cd continuous-wave laser, a monochromatic light from a 300 W Xenon lamp, and UV spectrometers (Maya2000, Ocean Optics, Orlando, FL, USA) as a PL detector at room temperature.

## 3. Results and Discussion

Figure 1 schematically shows the experimental process: the sample preparation, size separation, and measurement of the optical properties of WX_2_-*m*QDs. From the bulk WS_2_ (WSe_2_), the Na, K-tartrate intercalated composites were produced using a thermostatic oven and then radically dropped into the water with mild sonication to be broken into QDs with a certain size distribution. After the size separation via filtration and dialysis, the WX_2_-*m*QDs were re-dispersed in water with a pH of 2 and 11 for the measurement of their optical properties. More experimental details are provided in the Supporting Information. With the as-prepared WX_2_-*m*QDs, the structural properties of WX_2_-*m*QDs were examined at pH 7 and are referred to in the previous work [3]. Transmission electron microscopy (TEM) measurements were performed for the WX2-*m*QDs suspended in pH 7 using a lacy carbon TEM grid (Figure 2). The insets of Figure 2a,c are the digital images of dispersed WS_2_-*m*QDs and WSe_2_-*m*QDs, respectively. Figure 2b,d are TEM images of WS_2_-*m*QDs and WSe_2_-*m*QDs, respectively, with a lattice parameter. From TEM (Figure 2 and Ref. [3]) and AFM (Appendix A) analysis, the collected WX2-*m*QDs are revealed to not only have nano-sized structure, but also have two different sizes (WX_2_-*m*QDs-Ss with diameter of ~3 nm similar to the exciton Bohr radius; WX_2_-*m*QDs-Ls with diameter of ~12 nm, sufficiently larger than the exciton Bohr radius). The XPS spectra of the tungsten core level (W4f) (Appendix A) show four separated peaks, where the former two peaks at 32 and 34.2 eV correspond to the characteristic peaks of oxygen-free W4f and the latter at 35.2 and 37.5 eV originate from oxidized W4f [29]. The binding energies of the oxygen-free W4f peaks are ~1 eV lower than those from 2H WS_2_ and WSe_2_, indicating that the as-prepared WX_2_-*m*QDs have metallic phases (1T/1T’) even though the semiconducting phase cannot be totally excluded. The edges of the WX_2_-*m*QDs are modified by the hydroxide (OH^−^) and/or trivial tartrate (C_4_H_4_O_6_^2−^) groups during the exfoliation and fracturing process. The WX_2_-*m*QDs are observed to be negatively charged in the neutral pH by zeta potential (Appendix A). During the protonation or further oxidation, there is little possibility for further edge-functionalization except for H^+^ or OH^−^ because only NaOH or HCl solutions were used. The WX_2_-*m*QDs were well dispersed in the water with pH of 2 and 11 for hours and they were used in the further experiments without any treatment. However, the aggregated WX_2_-mQDs were also easily re-dispersed by mechanical and ultrasonic agitation even after days.

Figure 3 shows the UV-Vis absorbance spectra of WS_2_-*m*QDs-*S* (a), WS_2_-*m*QDs-*L* (b), WSe_2_-*m*QDs-*S* (c), and WSe_2_-*m*QDs-*L* (d), respectively. The as-prepared TMD-QDs clearly show monotonically decreasing UV-Vis absorbance spectra, indicating the metallic properties of the QDs [3]. In the protonated WX_2_-*m*QDs-*S*s (black in Figure 3a,c), a new absorption peak clearly appeared at ~4.2 eV (H-transition). This is in contrast to the quasi-continuously increasing absorbance spectra of the oxidized WX_2_-*m*QDs-*S*s (Figure 3a,c) and WX_2_-*m*QDs-*L*s (Figure 3b,d).

The H-transition band is much higher than the optical transition bands (~1.9 and ~2.3 eV) between the conduction band and the valance band as shown in the insets of Figure 3a,c [30,31]. Interestingly, the optical transition peak at ~2.1 eV in the protonated WX2-*m*QDs-*S*s is not observed in the WX_2_-*m*QDs-*L*s (inset Figure 3b,d). It should also be noted that the H-transition is a feature of only protonated WX_2_-*m*QDs-*S*s. The continuously increasing absorbance in the oxidized WX_2_-*m*QDs-*S*s and WX_2_-*m*QDs-*L*s was attributed to the overlap of high-energy excitonic absorptions and strong electron–phonon coupling [22]. However, we postulate in this work that the H-transition originates from the quantum confinement enhanced by the protonation of WX_2_-*m*QDs-*S*s because the protonation can suppress the edge effect of other functional groups and it supports the formation of highly localized excitonic states [32]. On the other hand, the oxidation induces the sub-energy states around the band edges of the WX_2_-*m*QDs-*S*s and WX_2_-*m*QDs-*L*s, resulting in a continuously increasing absorbance [15,33]. The effective band gap (*E_eff_*) in the QDs can be derived by Equation [11]:$$E_{eff}=E_{g}+\frac{h^{2}}{8m_{r}r^{2}}-\frac{1.8e^{2}}{4\varepsilon_{0}\varepsilon_{r}},$$
where *E_g_* is the band gap of bulk material, *h* is Plank’s constant, *m_r_* is the reduced mass of the exciton, *r* is the radius of the quantum dot, and *ε*_0_ (*ε_r_*) is the permittivity of free space and the relative permittivity of the material. Based on the size distribution of the WX_2_-*m*QDs (Appendix A), the bandgap distribution of WX_2_-*m*QDs-*S*s is 4~6 eV, whereas it is much lower than 2 eV in the WX_2_-*m*QDs-*L*s. The observed H-transition is located within the bandgap of the WX_2_-*m*QDs-*S*. The weak band around 2 eV is due to the size distribution of the WX_2_-*m*QDs-*S*. The absence of a UV-Vis absorbance peak in the WX_2_-*m*QDs-*L*s can be explained by the oxidation-induced sub-energy states and an *E_eff_* much lower than 2 eV.

Figure 4, Figure 5 and Appendix A show the comparison of PL spectroscopic results between the protonation and oxidation of WS_2_-*m*QDs-*S* and WSe_2_-*m*QDs-*S*. In Appendix A, the protonated WS_2_-*m*QDs-*S* show a harmonic PL (*h*PL) peak at ~2.95 eV and the full width at half maximum (FWHM) of ~0.78 eV, while in the oxidized WS_2_-*m*QDs-*S*, the *h*PL spectrum redshifts as much as ~0.03 eV (1%) and broadens the FWHM by about 0.17 eV (21.8%). The WSe_2_-*m*QDs-*S* show a similar trend in the *h*PL spectra, with a deviation of less than 5% (Appendix A). The excitation wavelength (λ_ex_)-dependent PL (PLE) spectra were measured to examine the change in the optical energy band structure in WX_2_-*m*QDs-*S*s and WX_2_-*m*QDs-*L*s according to the protonation and oxidation. Figure 4 shows the PL spectra of WS_2_-*m*QDs-*S* (a) and WSe_2_-*m*QDs-*S* (b), respectively, with a variation in λ_ex_ from 250 nm to 450 nm. In the protonated WX_2_-*m*QDs-*S*s, the PL peak at ~2.9 eV radically redshifts to ~2.4 eV when the λ_ex_ is larger than 370 nm. The intensity sharply decreases from λ_ex_ ~290 nm (~4.28 eV), which is around the H-transition band. On the other hand, the oxidized WX_2_-*m*QDs-*S*s show a gradual shift of the PL peak from ~2.9 eV at λ_ex_ ~250 nm to ~2.25 eV as the λ_ex_ increases up to 450 nm. Additionally, the spectra have a shoulder on the lower emission energy (~2.5 eV) with a broad FWHM. This indicates that although in the protonated WX_2_-*m*QDs-*S*s most excitons efficiently occur with high excitation energy decay at ~2.9 eV as the λ_ex_ is higher than 330 nm, in the oxidized WX_2_-*m*QDs-*S*s the correlation between the emission energy and the λ_ex_ becomes strong with the decreasing efficiency and another energy band for exciton decay.

Figure 5 shows the PL intensity of WS_2_-*m*QDs-*S* (a) and WSe_2_-*m*QDs-*S* (b), respectively, at a specific emission wavelength (λ_em_) as a function of the excitation energy. In the PL intensity vs. excitation energy (E_ex_) of the WX_2_-*m*QDs-*S*s (Figure 5), two effective E_ex_ bands (~4.96 eV and ~4.2 eV) are observed. Among them, the E_ex_~4.2 eV looks to be correlated with the H-transition of the UV-Vis spectra. The other E_ex_~4.96 eV being higher, by as much as ~0.76 eV, than the E_ex_~4.2 eV is most likely due to the balance band splitting [3,4] and/or the conduction band spin splitting [34]. This trend is similarly observed in the oxidized WX_2_-*m*QDs-*S*. However, the two E_ex_ bands are more broadened and the boundary between the two bands becomes vague. Moreover, the relative intensity at the lower E_ex_ band increases compared to that in the oxidized WX_2_-*m*QDs-*S*s. This suggests that the oxidation of the WX_2_-*m*QDs-*S*s considerably affects the exciton behaviors even though it does not overwhelm the quantum confinement. The measured PL properties are listed in Table 1.

Figure 6, Figure 7 and Appendix A show the PL spectroscopic results from the larger quantum dots, WS_2_-*m*QDs-*L* and WSe_2_-*m*QDs-*L*. The *h*PL peak and FWHM of protonated WS_2_-*m*QDs-*L* are ~2.83 eV and ~0.89 eV, respectively, which are ~0.12 eV (4%) redshifted and ~0.11 eV (14%) broadened compared with those from the protonated WS_2_-*m*QDs-*S* (Appendix A). This can be attributed to the size effect. On the other hand, the oxidized WS_2_-*m*QDs-*L* show a more redshifted peak (~2.63 eV) and a broadened FWHM (~1.06 eV). WSe_2_-*m*QDs-*L* shows a similar trend in the *h*PL spectrum to the WS_2_-*m*QDs-*L* (Appendix A) (see details in Table 1). Consequently, it is clear that the WX_2_-*m*QDs-*L*s show redshifts (0.12–0.14 eV) in the peak and broadness (0.09–0.11 eV) of the FWHM as the size increases from ~3 nm to ~12 nm. However, the variance in the *h*PL peak and the FWHM due to the protonation/oxidation in the WX_2_-*m*QDs-*L*s is 0.16–0.2 eV (5.6–7.1%) and 0.17–0.21 eV (17–25%), respectively.

This indicates that the pH variation causes more change in the *h*PL spectrum of the WX_2_-*m*QDs-*L*s compared to the size increment. From the PLE spectra (Figure 6), the peak in the protonated WX_2-_*m*QDs-*L*s appears at ~2.9 eV until λ_ex_~330 nm and then redshifts as the λ_ex_ increases, which is similar to that of the WX_2_-*m*QDs-*S*s. This implies that the intrinsic optical band of the WX_2_-*m*QDs is located at ~2.9 eV, demonstrating a large Stokes shift due to the strong exciton binding energy [35]. However, when the WX_2_-*m*QDs-*L*s are oxidized, the highest emission appears at ~2.5 eV and the emission peak at ~2.9 eV becomes a shoulder in the spectrum. This is in an inverse shape to that shown in the WX_2_-*m*QDs-*S*s (Figure 4) and the protonated WX_2_-*m*QDs-*L*s. Moreover, the peak at ~2.9 eV is obviously suppressed when λ_ex_ ≥ 370 nm. Interestingly, another dramatic change is observed in the PL intensity vs. E_ex_ (Figure 7). In the protonated WX_2_-*m*QDs-*L*s, the PL intensity varying with a maximum intensity at λ_ex_~4.9 eV is similar to that in the WX_2_-*m*QDs-*S*s as the E_ex_ changes from 5.5 eV to 2.5 eV. On the other hand, when oxidized, the PL intensity becomes flat with curves and the maximum intensity appears at a lower E_ex_ than 4 eV. This indicates that in the oxidized WX_2_-*m*QDs-*L*s the excited exciton states are spread and the oxidation more critically affects the optical energy band structure of the WX_2_-*m*QDs. The origin of the PLE peak shift is generally due to the size distribution of the *m*QDs or defects [3,4,36]. However, our above hypothesis is that the size increment and oxidation lead to the occurrence of sub-energy states between the conduction band and valance band. The formation of sub-energy states leads to the migration of excited charges into the lower energy states and helps the recombination of excitons. This is indirectly supported by the more redshifted and broadened PL, smoothly decreasing PL intensity, and flat PLE intensity spectra. In addition, the higher quantum yield in the oxidized WX_2_-*m*QDs-*S*s (4.9 for WSe_2_ and 4.1 for WSe_2_) than that (2.8 and 3.3) of the protonated ones directly proves our hypothesis (Table 1).

Figure 8 shows the normalized PL intensity and PL center as a function of measurement time in the protonated and oxidized WX_2_-*m*QDs-*S*s (Figure 8a,c) and WX_2_-*m*QDs-*L*s (Figure 8b,d). The total measurement time of the time-resolved PL (TRPL) was 42 ns with λ_ex_~266 nm (Appendix A). In the protonated WX_2_-*m*QDs-*S*, the PL intensity sharply decreases to less than 0.2 within 20 ns and the PL center shifts by as much ~0.1 eV from 3.0 eV, whereas the oxidized WX_2_-*m*QDs-*S*s show a relatively smooth decreasing intensity with a larger center shift (0.3 eV). This suggests that there is more exciton transition into the lower energy states following emissive decay in the oxidized WX2-mQDs-*S*s. The protonated WX_2_-*m*QDs-*L*s show a similar variance in the TRPL spectra to the protonated WX_2_-*m*QDs-*S*s with a ~0.1 eV lower PL center. However, in the oxidized WX_2_-*m*QDs-*L*s the PL center starts to appear at ~2.65 eV and then shifts only ~0.1 eV within the measurement time. The PL intensity decreases slightly slower than that of the protonated WX_2_-*m*QDs-*L*s. It should be noted that the average PL lifetimes of the oxidized WX_2_-*m*QDs-*L*s are 3.53 ± 0.11 ns for WS_2_-*m*QDs-*L* and 3.42 ± 0.11 ns for WSe_2_-*m*QDs-*L*, which are longer than those (2.99 ± 0.21 ns and 3.16 ± 0.14 ns) of the protonated ones (Appendix A). However, chalcogen dependence is not observed. This indicates that the exciton intraband transition quickly occurs within the broad exciton band above the band edge in the oxidized WX_2_-*m*QDs-*L*s.

So far, we have analyzed the UV-Vis, PL, PLE, and TRPL results measured on WX_2_-*m*QDs-*S*s and WX_2_-*m*QDs-*L*s according to protonation and oxidation. The solution with a pH of 2 can make the edge of WX_2_-*m*QDs protonated and even heal the chalcogen vacancy of the crystal structure. This might enhance the exciton localization by quantum confinement and suppress the occurrence of sub-energy states. Moreover, this protonation might lead to energy dissipation through the hydrogen bond interacting with the surrounding solvent molecules, e.g., H_2_O. On the other hand, oxidation-induced defect states appear around the top valance band and the lowest intrinsic conduction band. Figure 9 is schematically drawn for the optical energy band structures of the protonated WX_2_-*m*QDs-*S*s and the oxidized WX_2_-*m*QDs-*L*s. In the protonated WX_2_-*m*QDs-*S*s, there are discrete energy states due to the enhanced quantum confinement that causes the localization of charges. On the other hand, the oxidized WX_2_-*m*QDs-*L*s have a broad exciton band above the band edge resulting in exciton delocalization and there are also extra sub-energy states under the conducting band and over the valance band [4]. However, the simple increment in size might not be critical and rather have a negative effect on the quantum yield of WX_2_-*m*QDs. The decisive factor for quantum confinement is the exciton Bohr radius (*R*), which can be deduced by the following equation:$$R=\varepsilon \left(\frac{m_{0}}{m_{r}}\right)a_{0}$$
where *ε* is the dielectric constant of the material, *m*_0_ is the free electron mass, *m_r_* is the reduced mass of the exciton, and *a*_0_ ≈ 0.5 Å. Based on the parameters in previous reports [17,18], the *R* of WX_2_-*m*QDs is about 3~4 nm. This suggests that when the size of WX_2_-*m*QDs is comparable to the *R*, the quantum confinement more dominantly affects the PL properties than the environmental effect, whereas when the size >>*R* (for example, the WX_2_-*m*QDs-*L*s) the protonation/oxidation plays an important role in the photophysical properties. However, there must be a transition point of the dominant mechanism between the quantum confinement and the environmental effect as the size increases, and this question still remains elusive.

## 4. Conclusions

We investigated the effect of protonation and oxidation on the photophysical properties of two different size WX_2_-*m*QDs. The protonated WX_2_-*m*QDs showed relatively sharp harmonic PL spectra with a high emission energy peak, two obvious PLE intensity peaks, short lifetimes, and low quantum yield. On the other hand, the oxidized ones showed relatively broad and lower harmonic PL energy peaks with longer lifetimes and higher QYs. In the WX_2_-*m*QDs-*L*s, the variation in PL properties was dominantly affected by protonation/oxidation rather than the size effect. This demonstrates that in the WX_2_-*m*QDs with a larger size than the exciton Bohr radius, the photophysical properties are more effectively affected by the environmental conditions, whereas in the WX_2_-*m*QDs with a size comparable to the exciton Bohr radius, the quantum confinement is a dominant mechanism that explains the optical properties. This work illuminates the transition of the dominant mechanism modulating the exciton dynamics in WX_2_-*m*QDs and it is expected to be an important guide for the practical application of TMDs-*m*QDs. However, we still need to investigate the exact size at which the transition from quantum confinement to the environmental effect occurs.

## Figures and Tables

**Figure 1 nanomaterials-13-02075-f001:**
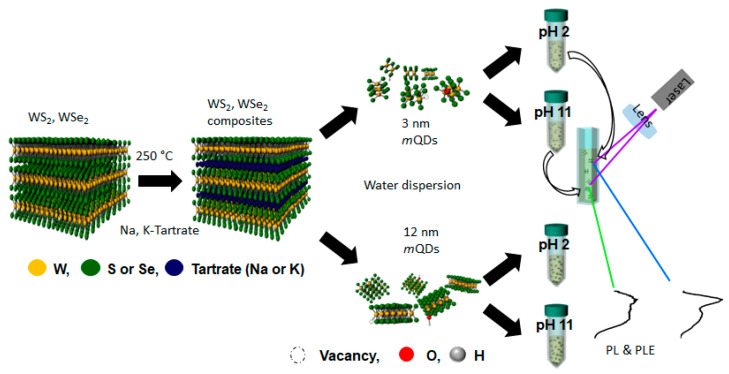
Schematically shown synthesis process of WS_2_-*m*QDs-*S* and WS_2_-*m*QDs-*L*.

**Figure 2 nanomaterials-13-02075-f002:**
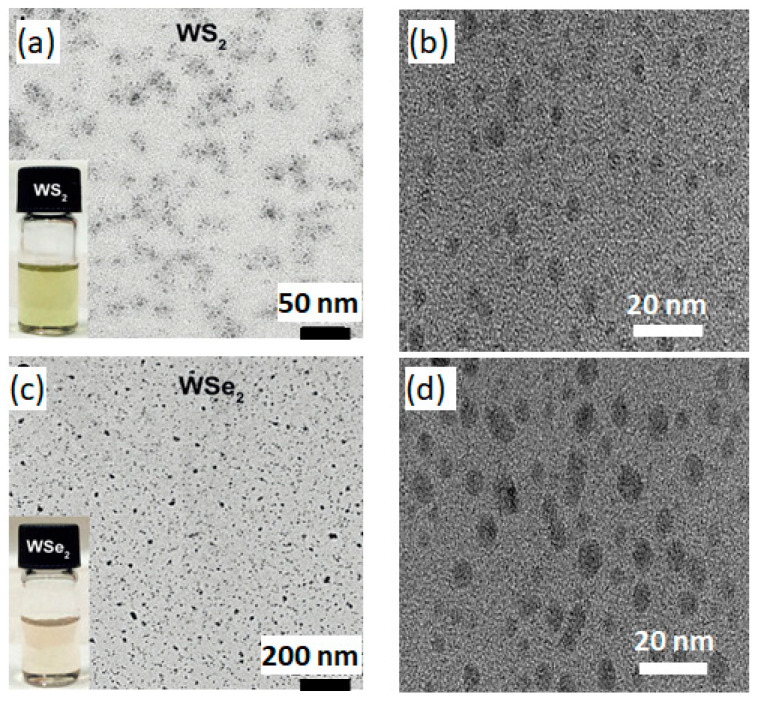
TEM images of WS_2_-*m*QDs (**a**,**b**) and WSe_2_-*m*QDs (**c**,**d**).

**Figure 3 nanomaterials-13-02075-f003:**
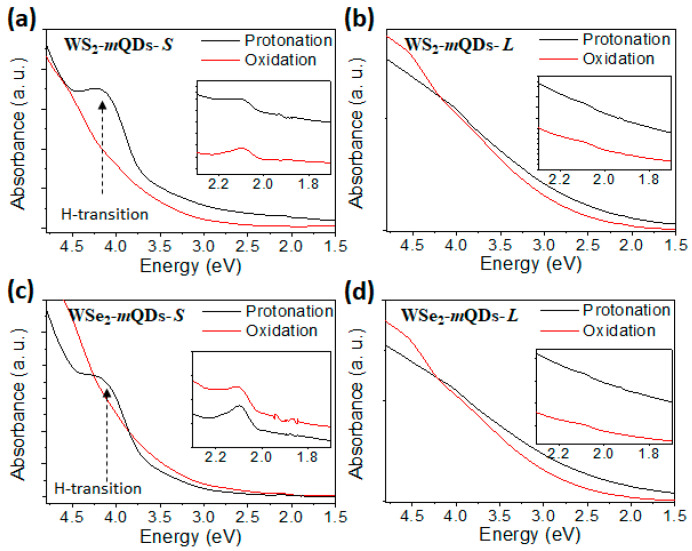
UV-Vis absorbance of WX_2_-*m*QDs-*S*s and WX_2_-*m*QDs-*L*s. UV-Vis absorbance of WS_2_-*m*QDs-*S* (**a**), WS_2_-*m*QDs-*L* (**b**), WSe_2_-*m*QDs-*S* (**c**), and WSe_2_-*m*QDs-*L* (**d**). Insets: expansion of lower wavelength range.

**Figure 4 nanomaterials-13-02075-f004:**
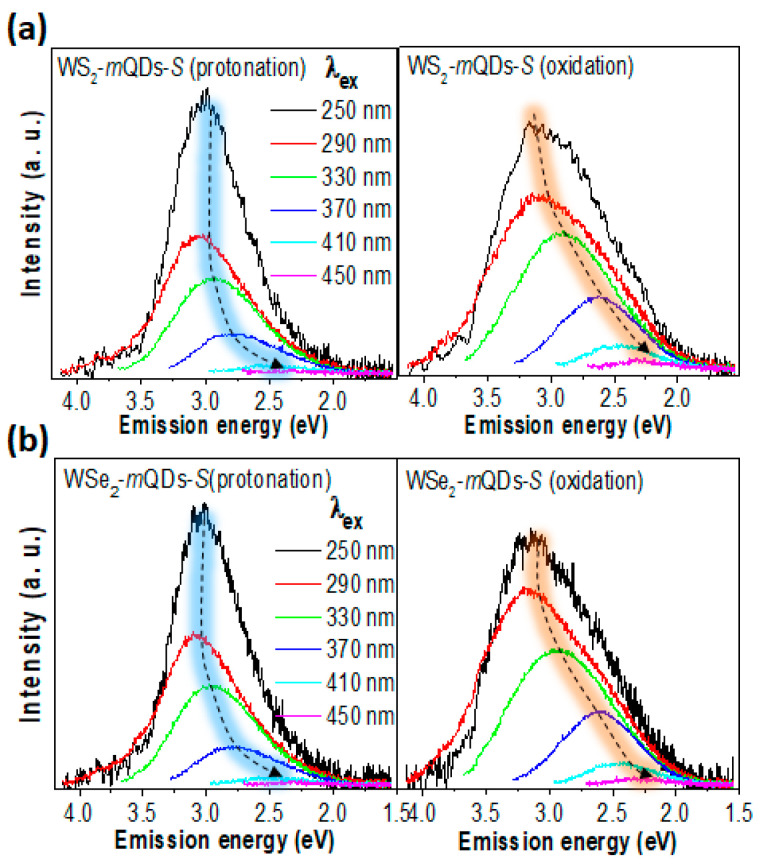
PLE spectra of WX_2_-*m*QDs-*S*s. PLE spectra of protonated (left) and oxidized (right) WS_2_-*m*QDs-*S* (**a**) and WSe_2_-*m*QDs-*S* (**b**) measured at λ_ex_ from 250 nm to 450 nm.

**Figure 5 nanomaterials-13-02075-f005:**
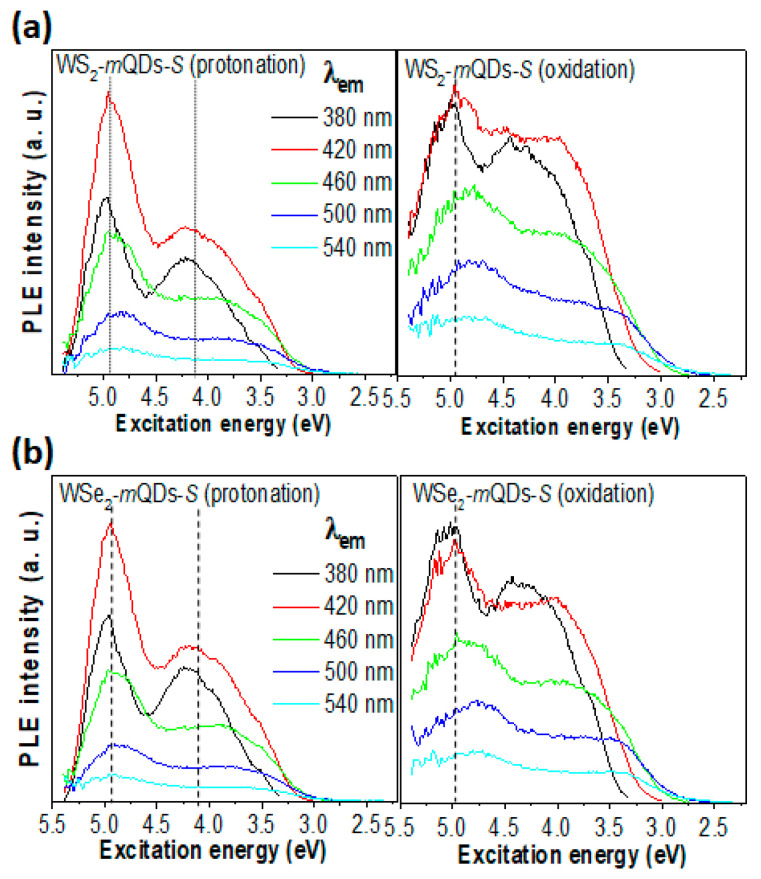
PLE intensity vs. excitation energy of WS_2_-*m*QDs-*S* (**a**) and WSe_2_-*m*QDs-*S* (**b**) at emission wavelengths from 380 to 540 nm. The left panel is for protonation and the right is for oxidation.

**Figure 6 nanomaterials-13-02075-f006:**
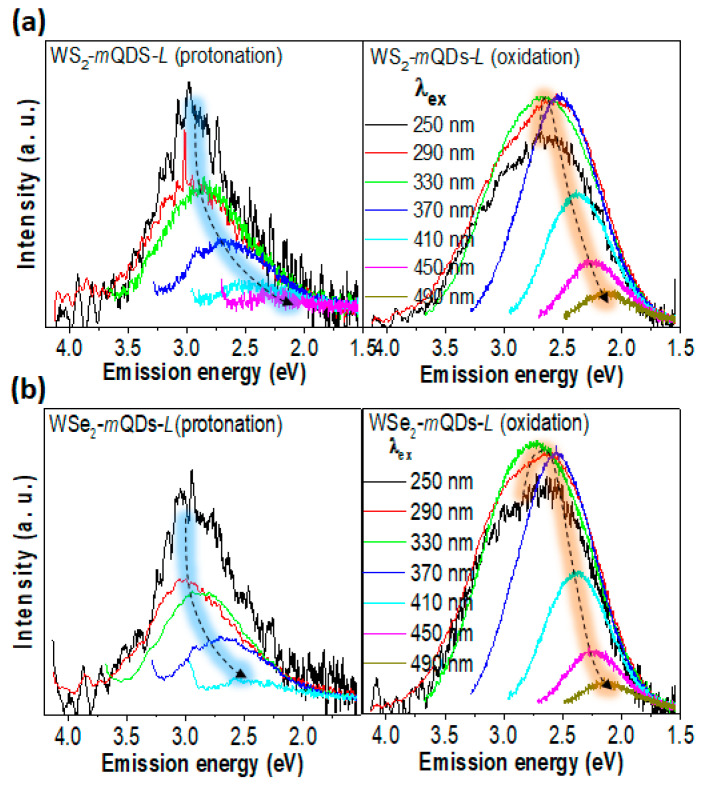
PLE spectra of WX_2_-*m*QDs-*L*s. PLE spectra of protonated (left) and oxidized (right) WS_2_-*m*QDs-*S* (**a**) and WSe_2_-*m*QDs-*S* (**b**) measured at λ_ex_ from 250 nm to 450 nm.

**Figure 7 nanomaterials-13-02075-f007:**
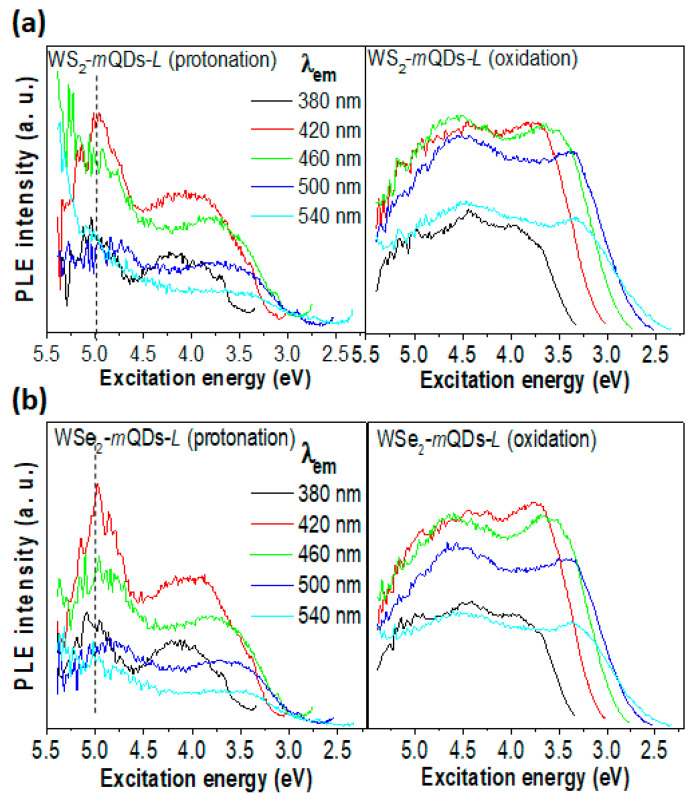
PLE intensity vs. excitation energy of WS_2_-*m*QDs-*L* (**a**) and WSe_2_-*m*QDs-*L* (**b**) at emission wavelengths from 380 to 540 nm. The left panel is for protonation and the right is for oxidation.

**Figure 8 nanomaterials-13-02075-f008:**
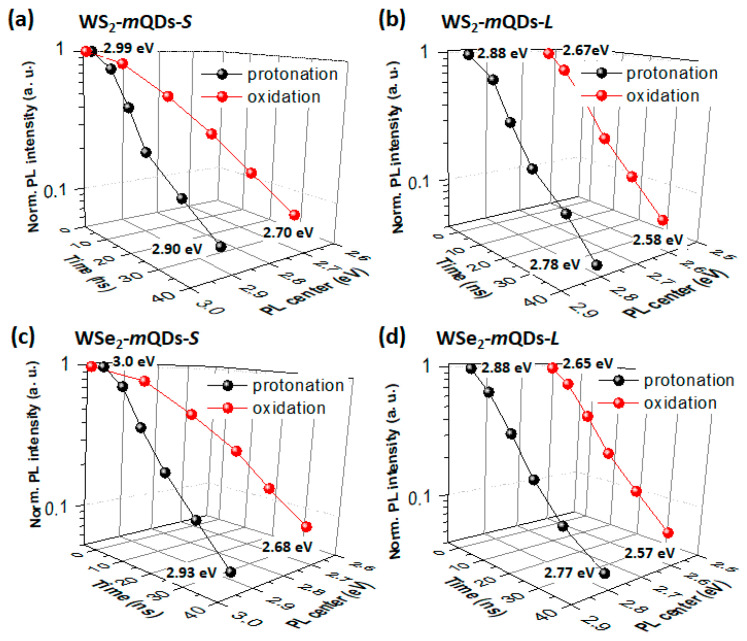
Time-resolved PL intensity vs. PL peak center vs. measurement delay time. The long scaling normalized PL intensity and PL peak shift as a function of measurement time delay in WS_2_-*m*QD-*S* (**a**), WS_2_-*m*QD-*L* (**b**), WSe_2_-*m*QD-*S* (**c**), WSe_2_-*m*QD-*L* (**d**), respectively, with λ_ex_~266 nm. Each point has a PL accumulation time of 0–2, 2–6, 6–12, 12–20, 20–30, and 30–42 ns.

**Figure 9 nanomaterials-13-02075-f009:**
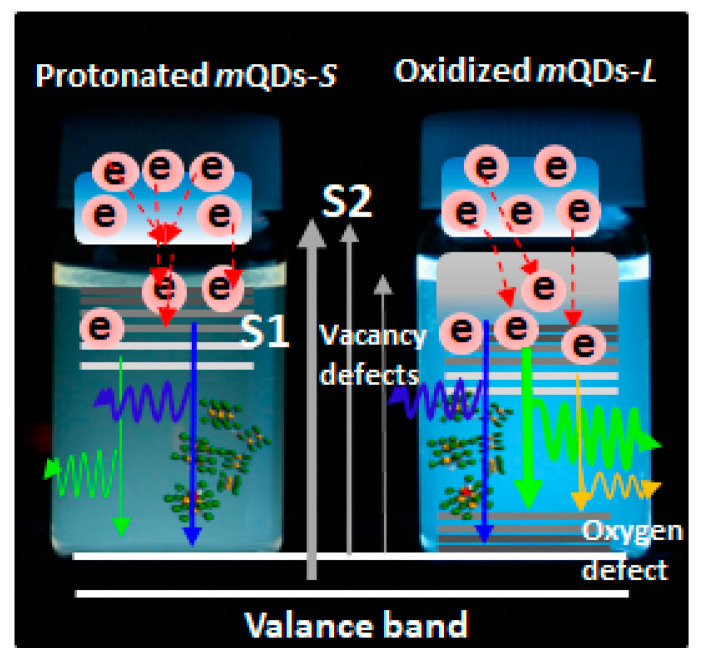
Schematically shown photoelectron dynamics in WX_2_-*m*QD-*S*s and WX_2_-*m*QD-*L*s.

**Table 1 nanomaterials-13-02075-t001:** Photophysical parameters of WS_2_ (Se_2_)-*m*QDs.

Materials	pH	PL Peak (eV)	FWHM (eV)	QY (%)	Materials	pH	PL Peak (eV)	FWHM (eV)	QY (%)
WS_2_-*m*QDs-*S*	2	2.95	0.78	2.8	WSe_2_-*m*QDs-*S*	2	2.98	0.78	3.3
	11	2.92	0.95	4.9		11	2.93	0.97	4.1
WS_2_-*m*QDs-*L*	2	2.83	0.89		Wse_2_-*m*QDs-*L*	2	2.84	0.87	
	11	2.63	1.06			11	2.68	1.09	

## Data Availability

Not applicable.

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
