# Peer review of "Competitive Effects of Oxidation and Quantum Confinement on Modulation of the Photophysical Properties of Metallic-Phase Tungsten Dichalcogenide Quantum Dots"

_nanomaterials, 2023, doi:10.3390/nano13142075_

Round 1

Reviewer 1 Report

This manuscript demonstrate that the oxidation effect prevails the quantum confinement on the metallic phase tungsten dichalcogenides WX2-mQDs when the QD size becomes larger than the exciton Bohr radius. The results is interesting, but it still has the following shortcomings. Thus, I recommend this work can be published in the Nanomaterials after major revisions.

1. As the size of the WX2 larger than the exciton Bohr radius, whether it can still be called quantum dots?

2. The Figures is fuzzy, it is suggested to replace the HD Figures.

3. The TEM images of the QDs in the Supporting Information are suggested to transfer to the main body.

4. There are so many elements in Figure 4e, that the authors is suggest to simplify it to highlight the point of the diagram.

 Minor editing of English language required.

Author Response

# Reviewer: 1

Comments: This manuscript demonstrate that the oxidation effect prevails the quantum confinement on the metallic phase tungsten dichalcogenides WX2-mQDs when the QD size becomes larger than the exciton Bohr radius. The results is interesting, but it still has the following shortcomings. Thus, I recommend this work can be published in the Nanomaterials after major revisions.

The authors report a study on the effects of quantum confinement and of oxidation/protonation on the optical properties of metallic tungsten dichalcogenide quantum dots. They study two sizes of WS2 and WSe2 QDs (3nm and 12nm) and characterize the photophysics including absorption and photoluminescence spectra, photoluminescence excitation, and time resolved PL (and quantum yield for the 3nm QDs). The main results are that quantum confinement dominates for the small QD (size around or below the exciton Bohr radius) – although I note that edge effects are still there as can be seen from the reduced confinement for oxidised QDs. For bigger size (12nm) the protonation/oxidation has a stronger effect.

The study and its results will be I think of interest to the community. I have however a certain number of comments/questions (see below). I’d ask the authors to clarify these points before a decision to publish can be made.

First of all, the authors appreciate reviewer’s valuable comments and questions. Based on them, we have confidently revised and modified the manuscript.

  1. As the size of the WX2 larger than the exciton Bohr radius, whether it can still be called quantum dots?

Response for question 1

Thank you for the question. The reviewer’s question for the critical size of quantum dots is non-trivial question. It is typically referred the quantum dots to the nanoparticles exhibiting quantum confinement effects. When the size of particle become small and small to be comparable or smaller than the exciton Bohr radius, the electronic and optical properties of materials start to be quite different from those of bulk. However, if the size is larger than the exciton Bohr radius, this quantum effects become less clear. However, in most literature the ‘quantum dots’ are used more broadly to refer the particles exhibiting size-dependent properties even though their size is larger than the exciton Bohr radius. In this point, the WX2 particles larger than the exciton Bohr radius can be referred to quantum dots as they show size dependent properties.

  1. The Figures is fuzzy, it is suggested to replace the HD Figures.

Response for question 2

Thank you for the valuable comment. According to the reviewer’s comment, we revised and reorganized the Figures. Furthermore, the main manuscript has nine Figures with corresponding text and figure caption to increase understanding. This is believed to be proper response of reviewer’s comment.

  1. The TEM images of the QDs in the Supporting Information are suggested to transfer to the main body.

Response for question 3

According to the reviewer’s comment, we moved the TEM images into the main text as Figure 2 as following.

Figure 2. TEM images of WS2-mQDs (a and b) and WSe2-mQDs (c and d).

  1. There are so many elements in Figure 4e, that the authors are suggest to simplify it to highlight the point of the diagram.

Response for question 4

Thank you for your comment. Actually, Figure 4e looked to be complicated when it was a part of Figure 4. To enhance understanding, we divided the previous Figure 4 into two Figures and changed Figure 4e into Figure 9. Overall, the paper was modified by dividing 4 figures into 9 figures.

  1. Comments on the Quality of English Language: Minor editing of English language required.

Response for question 5

According to the reviewer’s comment, we had a proofreading from professional native-speaker and revised the manuscript. We attached the confirmation file.

Reviewer 2 Report

Dear Editor, dear authors,

Please find here my review of the manuscript entitled ‘Competitive effects of oxidation and quantum confinement on the modulation of photophysical properties of the metallic  phase tungsten dichalcogenides quantum dots’ by Bo-Hyun Kim et al.

The authors report a study on the effects of quantum confinement and of oxidation/protonation on the optical properties of metallic tungsten dichalcogenide quantum dots. They study two sizes of WS2 and WSe2 QDs (3nm and 12nm) and characterize the photophysics including absorption and photoluminescence spectra, photoluminescence excitation, and time resolved PL (and quantum yield for the 3nm QDs). The main results are that quantum confinement dominates for the small QD (size around or below the exciton Bohr radius) – although I note that edge effects are still there as can be seen from the reduced confinement for oxidised QDs. For bigger size (12nm) the protonation/oxidation has a stronger effect.

The study and its results will be I think of interest to the community. I have however a certain number of comments/questions (see below). I’d ask the authors to clarify these points before a decision to publish can be made.

·   The quality of the figures is poor – they appear blurry. This needs to be addressed throughout.

·       Correct the term valence (and not valance) throughout.

·       Materials and methods:

o   Clarify how the 2 size of QDs are made;

o   Indicate the source of the materials (e.g. suppliers of WSe)

o   Give full details of characterization set-ups (including PL and PLE, PLQY and TRPL)

·       Table 1: I would add the average PL lifetimes of the materials in this table. Also, the PLQY for the large QD should be included, or else justify why it is not.

·       Define all acronyms the first time they are use – throughout the manuscript.

·       Line 87 to 93: remove that block of text.

·       Explain the naming for the H-transition.

·       Line 158/159: explain that statement. I am not sure I understand how the size distribution has a role in the absorption feature seen at around 2eV.

·       It would be nice to have more discussion justifying the metallic nature of these QDs.

·       Equation:

o   The number is missing

o   I think the radius should appear in the third term on the right-hand side (the term describing Coulomb interaction)

·       What is the meaning of harmonic in ‘harmonic PL’?

·       I am not convinced by the claim that in the small QDs the quantum confinement dominates (even if they are present) over edge/surface effects; indeed, the oxidized QD do not show a pronounced H-transition. Can the authors comment and/or add a justification?

·       Why do the oxidized QDs have a higher PLQY?

·       Justify or reformulate the statement (line 250) ‘This implies that there is more exciton interband transition during emissive decay in the oxidized WX2-mQDs-S.’

·       Clarify the sentences lines 259 and 260. Why does it indicate a fast transition within the exciton band above the band edge? Maybe reformulate the sentence as it is not clear.

·       Fig. 4: these are not clear. 2D plots would probably be better for these; Fig 4e is not that clear either – it needs to be explained/detailed in the text. Some of this information may be in in ref [3] but should appear here a s well.

·       Please read and review/edit the grammar throughout, which should clarify some of the points raised above for the reader.

 Best wishes

The English is overall ok. I recommend reviewing/editing the grammar throughout as needed, which should clarify some aspects of the discussion. 

Author Response

# Reviewer: 2

Comments:   1. The quality of the figures is poor – they appear blurry. This needs to be addressed throughout.

Response for question 1

Thank you for your suggestion. According to the reviewer’s comment, we revised the Figures from 4 figures to 9 figures for the enhancement of visibility.

2. Figure S1. The histogram insert is unreadable, and the font is too small and blurry to read the size distribution. 

Response for question 2

Thank you for your comment. According to the comment, Fig. S1 was modified as following.

3. Correct the term valence (and not valance) throughout.

Response for question 3

Thank you for your comment. According to the request of reviewer, we corrected the typo throughout.

  1. Clarify how the 2 size of QDs are made;

Response for question 4

Thank you for your comment. For the size separation method of TMD-QDs, we already mentioned it via references 5 and 18. Furthermore, we revised in accordance the reviewer’s comments the Materials and Methods parts in more detail in manuscript.

Before: “Briefly, the initial steps start with the mixing and grinding of potassium sodium tartrate (200 mg) with WS2 (WSe2) (20 mg). Then, the ground homogeneous mixtures are reacted in an autoclave vessel at 250 ℃ for 12 h, and then instantly exfoliated in water with sonication. The sizes of the WX2-mQDs were firstly controlled by AAO (20 nm) filtration and then dialyzed using dialysis tubing, which simultaneously removed the remaining salts.”

After: “Briefly, the initial steps start with the mixing and grinding of potassium sodium tartrate (200 mg) with WS2 (WSe2) (20 mg) purchased from Sigma Aldrich in Korea. Then, the ground homogeneous mixtures are reacted in an autoclave vessel at 250 ℃ for 12 h, and then instantly exfoliated in water with sonication. The sizes of the WX2-mQDs were firstly controlled by AAO (20 nm, Whatman, Merch KGaA, Korea) filtration and then dialyzed using dialysis tubing (10,000 and 8,000 NMWL, Amicon Ultra-15, Merck KGaA, Korea) to separate into two sizes, which simultaneously removed the remaining salts.”

  1. Indicate the source of the materials (e.g. suppliers of WSe)

Response for question 5

Thank you for your comment. We revised in accordance the reviewer’s comments the Materials and Methods parts in more detail in manuscript.

Before: “Briefly, the initial steps start with the mixing and grinding of potassium sodium tartrate (200 mg) with WS2 (WSe2) (20 mg). Then, the ground homogeneous mixtures are reacted in an autoclave vessel at 250 ℃ for 12 h, and then instantly exfoliated in water with sonication. The sizes of the WX2-mQDs were firstly controlled by AAO (20 nm) filtration and then dialyzed using dialysis tubing, which simultaneously removed the remaining salts.”

After: “Briefly, the initial steps start with the mixing and grinding of potassium sodium tartrate (200 mg) with WS2 (WSe2) (20 mg) purchased from Sigma Aldrich in Korea. Then, the ground homogeneous mixtures are reacted in an autoclave vessel at 250 ℃ for 12 h, and then instantly exfoliated in water with sonication. The sizes of the WX2-mQDs were firstly controlled by AAO (20 nm, Whatman, Merch KGaA, Korea) filtration and then dialyzed using dialysis tubing (10,000 and 8,000 NMWL, Amicon Ultra-15, Merck KGaA, Korea) to separate into two sizes, which simultaneously removed the remaining salts.”

  1. Give full details of characterization set-ups (including PL and PLE, PLQY and TRPL)

Response for question 6

Thank you for your suggestion. According to the reviewer’s comment, we added details of characterization set-ups as following. Before: The morphologies of the mQDs were analyzed using an AFM in tapping mode under ambient conditions. XPS (Sigma Probe, AlKα), UV/Vis spectra, fluorescence spectra (Perkin-Elmer LS 55 luminescence spectrometer), and transmission electron microscopy (TEM, Titan cubed G2 60-300) analyses were conducted. PL measurements were carried out using a 325 nm He-Cd continuous-wave laser, a monochromatic light from a 300 W Xenon lamp, and UV spectrometers (Maya2000, Ocean Optics, USA) as a PL detector at room temperature. After: The morphologies of the mQDs were analyzed using an AFM (X2-70, Park Systems Corp, Korea,) in tapping mode under ambient conditions in the Smart Nature Research Centre. XPS (Sigma Probe, AlKα), UV/Vis spectra (UV-3600, Shimadzu spectrometer, Smart Nature Research Centre), fluorescence spectra (Perkin-Elmer LS 55 luminescence spectrometer), and transmission electron microscopy (TEM, Titan cubed G2 60-300) analyses were conducted. PL measurements were carried out using a 325 nm He-Cd continuous-wave laser, a monochromatic light from a 300 W Xenon lamp, and UV spectrometers (Maya2000, Ocean Optics, USA) as a PL detector at room temperature.

  1. Table 1: I would add the average PL lifetimes of the materials in this table. Also, the PLQY for the large QD should be included, or else justify why it is not.

Response for question 7

Thank you for your valuable comment. According to the reviewer’s comment, the average PL lifetimes were included in the table 1. However, the PLQY of the large QD could not be evaluated due to the low quantum efficiency. However, future research is needed to improve the quantum efficiency of QDs.

  1. Define all acronyms the first time they are use – throughout the manuscript.

Response for question 8

Thank you for your suggestion. According to the reviewer comment, we defined all acronyms the first time they were used.

  1. Line 87 to 93: remove that block of text.

Response for question 9

Thank you for your comment.  We removed the block of text.

  1.   Explain the naming for the H-transition.

Response for question 10

Thank you for your comment. In our manuscript, we identified an absorption peak around 4.2 eV in the manuscript, which is the highest energy band that can be observed with meaningful analysis in UV-Vis spectroscopy. So, we named it to be H-transition.

  1. Line 158/159: explain that statement. I am not sure I understand how the size distribution has a role in the absorption feature seen at around 2 eV.

Response for question 11

Thank you for your question. Line 158/159 is “The weak band around 2 eV is due to the size distribution of the WX2-mQDs-S.” This sentence is for the well-known UV-Vis absorption spectrum continuously increasing in the lower energy range due to the variation of particle size. In the graphene quantum dot and transition metal dichalcogendie quantum dots studies, it is usually observed the gradually increasing absorption spectra and it is explained by size distribution of QD particles. Based on those results, we ascribed the weak band around 2 eV to the size distribution the WX2-mQDs-S. Also, we have attached relevant references in this manuscript. (ref. 3, 21, 23, and 24)

  1. It would be nice to have more discussion justifying the metallic nature of these QDs.

Response for question 12

Thank you for your valuable comments. Actually, the metallic nature of these QDs were clearly probed and discussed in the previous report ‘Ref. 3, NPG Asia Mater. 2021, 13, 41.’.

Therefore, we added the sentence on line 2 on page 4 as shown below.

“The as-prepared TMD-QDs clearly shows monotonically decreasing UV-vis absorbance spectra, indicating the metallic properties of the QDs [3].”

  1. Equation:

The number is missing. And I think the radius should appear in the third term on the right-hand side (the term describing Coulomb interaction)

Response for question 13

Thank you for your comment. The equations using in manuscript are two and the numbering of equation is not mandatory. We don’t think the number of equations should be included for understanding what this manuscript wants to describe. In addition, the exciton Bohr radius of the particle is mentioned as “Based on the parameters in previous reports [17, 18], the R of WX2-mQDs is about 3 ~ 4 nm.”, which is comparable to the size of WX2-mQDs-S (~3 nm), but much smaller than that of WX2-mQDs-L (~12 nm) mentioned in the first paragraph of Results and Discussion.

  1. What is the meaning of harmonic in ‘harmonic PL’?

Response for question 14

Thank you for your question. A molecular system usually exhibits a broad PL spectrum because the excited electron in the molecule can decay by transitions of the ground electronic state to different vibrational and rotational states. This is why the PL spectrum shows several peaks or is divided into several sub-peaks. Also, excited molecules can be affected by different environments shifting the energy of both excited and ground states. While strong vibrational-electronic coupling induces several sharp peaks, solvent effects or side and edge effects tend to make a broader spectrum with less well-defined peaks. In our WX2-mQDs, the PL peaks are broadened with a FWHM of 0.7~1.1 eV, indicating that there may be lots of sub-peaks originated from different energy states. Therefore, in our manuscript, the harmonic PL means that the several sub-peaks harmoniously form one PL peak.

  1. I am not convinced by the claim that in the small QDs the quantum confinement dominates (even if they are present) over edge/surface effects; indeed, the oxidized QD do not show a pronounced H-transition. Can the authors comment and/or add a justification?

Response for question 15

Thank for your valuable suggestion. According to the reviewer comment, we discussed in the main text based on the experimental results.

In the revised manuscript,

Page 5: Based on the size distribution of the WX2-mQDs (Figure S1), the bandgap distribution of WX2-mQDs-S is 4 ~ 6 eV, whereas it is much lower than 2 eV in the WX2-mQDs-L. The observed H-transition is located within the bandgap of the WX2-mQDs-S. The weak band around 2 eV is due to the size distribution of the WX2-mQDs-S. The absence of a UV-Vis absorbance peak in WX2-mQDs-L can be explained by the oxidation induced sub-energy states and an Eeff much lower than 2 eV.

Although we partially conceive an agreement for reviewer’s comment, as we demonstrated in Page 4: Interestingly, the optical transition peak at ~ 2.1 eV in the protonated WX2-mQDs-S is not observed in WX2-mQDs-L (Inset Figure 3c and e). It should also be noted that the H-transition is a feature of only protonated WX2-mQDs-S. The continuous increasing absorbance in the oxidized WX2-mQDs-S and WX2-mQDs-L was attributed to the overlap of high-energy excitonic absorptions and strong electron-phonon coupling [18]. However, we postulate in this work that the H-transition originates from the quantum confinement enhanced by the protonation of WX2-mQDs-S because the protonation can suppress the edge effect of other functional groups and it supports the formation of high-localized excitonic states [28].’,  because the WX2-mQDs-S show less dependence of edge structure compared with WX2-mQDs-L, we believe our postulate that the quantum confinement effect dominates the environment effect in the WX2-mQDs-S.

  1. Why do the oxidized QDs have a higher PLQY?

Response for question 16

 Thank for your question. In the main text, we briefly discussed this point as following: The solution of pH 2 can make the edge of WX2-mQDs protonated and even heal the chalcogen vacancy of the crystal structure. This might enhance the exciton localization by quantum confinement and suppress the occurrence of sub energy states. Moreover, this protonation might lead to energy dissipation through the hydrogen bonding interacting with the surrounding solvent molecules, e.g., H2O. On the other hand, oxidation induced defect states appear around the top valance band and the lowest intrinsic conduction band.

  1. Justify or reformulate the statement (line 250) ‘This implies that there is more exciton interband transition during emissive decay in the oxidized WX2-mQDs-S.

Response for question 17

Thank you for your kind comment. This makes us seriously reconsider this description. Because we think we need more study to justify this statement, we finally reformulate the statement as following.

Before: This implies that there is more exciton interband transition during emissive decay in the oxidized WX2-mQDs-S.

After: This implies that there is more exciton transition into the lower energy states following emissive decay in the oxidized WX2-mQDs-S.

  1. Clarify the sentences lines 259 and 260. Why does it indicate a fast transition within the exciton band above the band edge? Maybe reformulate the sentence as it is not clear.

Response for question 18

Thank you for your comment. Because we think we need more study to justify this statement, we finally reformulate the statement as following.

Before: This implies that there is more exciton interband transition during emissive decay in the oxidized WX2-mQDs-S.

After: This implies that there is more exciton transition into the lower energy states following emissive decay in the oxidized WX2-mQDs-S.

  1. Fig. 4: these are not clear. 2D plots would probably be better for these; Fig 4e is not that clear either – it needs to be explained/detailed in the text. Some of this information may be in in ref [3] but should appear here as well.

Response for question 19

Thank you for your comment. Actually, figure 4e looked to be complicated when it was a part of Figure 4. To enhance understanding, we divided the previous Figure 4 into two Figures and changed Figure 4e into Figure 9. Overall, the paper was modified by dividing 4 figures into 9 figures.

Reviewer 3 Report

The text of the manuscript under review is completely unreadable. There are many grammatical and semantic errors. English is incomprehensible. I think that the manuscript does not meet the high standards of the journal Nanomaterials and should be rejected.

English is incomprehensible.

Author Response

# Reviewer: 3

Comments:  The text of the manuscript under review is completely unreadable. There are many grammatical and semantic errors. English is incomprehensible. I think that the manuscript does not meet the high standards of the journal Nanomaterials and should be rejected.

Response for Comment

Authors are sorry for the language problem as mentioned by Reviewer. To improve English in the manuscript, the manuscript was carefully reviewed again by authors and had a proofreading from a native speaker and professional editing service in KAIST. All revised parts were reflected in the main text. Furthermore, to enhance understanding, the paper was modified by dividing 4 figures into 9 figures and the content was reinforced by reflecting the opinions of all reviewers.

Round 2

Reviewer 3 Report

No doubt the Authors have done their best to improve the manuscript thoroughly based on all the comments indicated by the referees. The revised version of the manuscript is a completed and well-organized research paper. I only recommend Authors fix minor bugs to improve the article.

A) Add some recent references on QDs applications:

1. Vitukhnovsky A.G., Zvagelsky R.D., Kolymagin D.A., Pisarenko A.V., Chubich D.A., Three-dimensional optical lithography and nanoscale optical connectors, Bulletin of the Russian Academy of Sciences: Physics, 84(7), 760 (2020). DOI: 10.3103/S1062873820070321.

2. Mohammad Ali Farzin, Hassan Abdoos. A critical review on quantum dots: From synthesis toward applications in electrochemical biosensors for determination of disease-related biomolecules, Talanta, 224, 121828 (2021). DOI: 10.1016/j.talanta.2020.121828.

3. Nayab Azam, Murtaza Najabat Ali, Tooba Javaid Khan. Carbon Quantum Dots for Biomedical Applications: Review and Analysis, Frontiers in Materials, 8, 700403 (2021). DOI: 10.3389/fmats.2021.700403.

4. A.I. Arzhanov, A.O. Savostianov, K.A. Magaryan, K.R. Karimullin, A.V. Naumov. Photonics of semiconductor quantum dots: applied aspects, Photonics Russia, 16(2) 96 (2022). DOI: 10.22184/1993-7296.FRos.2022.16.2.96.112.

B) Correct some misprints in the text, e.g. “…oxidation effect prevails overthe quantum confinement…” – “over the” (Abstract, line 4); “32. 32…” in the list of references etc.

C) Φ should be replaced by “diameter” or D (page 2, line 19; page 3, lines 15 and 16).

D) Figures 2-9 should be increased by the size and centered.

The article can be accepted for publication in Nanomaterials after this minor revision.

Author Response

# Reviewer: 3

Comments: No doubt the Authors have done their best to improve the manuscript thoroughly based on all the comments indicated by the referees. The revised version of the manuscript is a completed and well-organized research paper. I only recommend Authors fix minor bugs to improve the article.

  1. A) Add some recent references on QDs applications:
  2. Vitukhnovsky A.G., Zvagelsky R.D., Kolymagin D.A., Pisarenko A.V., Chubich D.A., Three-dimensional optical lithography and nanoscale optical connectors, Bulletin of the Russian Academy of Sciences: Physics, 84(7), 760 (2020). DOI: 10.3103/S1062873820070321.
  3. Mohammad Ali Farzin, Hassan Abdoos. A critical review on quantum dots: From synthesis toward applications in electrochemical biosensors for determination of disease-related biomolecules, Talanta, 224, 121828 (2021). DOI: 10.1016/j.talanta.2020.121828.
  4. Nayab Azam, Murtaza Najabat Ali, Tooba Javaid Khan. Carbon Quantum Dots for Biomedical Applications: Review and Analysis, Frontiers in Materials, 8, 700403 (2021). DOI: 10.3389/fmats.2021.700403.
  5. A.I. Arzhanov, A.O. Savostianov, K.A. Magaryan, K.R. Karimullin, A.V. Naumov. Photonics of semiconductor quantum dots: applied aspects, Photonics Russia, 16(2) 96 (2022). DOI: 10.22184/1993-7296.FRos.2022.16.2.96.112.

Response for question A

Thank you for your invaluable comments. According to your comments, the references for the recent works on QD applications were added in introduction part of the revised manuscript.

Added references in the revised manuscript

Modulating the optical properties of semiconducting TMDs-QDs has been conducted by using size control [3, 4, 10, 11,12], environmental solvent [13-15], surface/edge func-tionalization [16], defect control [17], heterostructure [18], and structural phase control [5].  These modulation techniques have found applications across various fields [19,20].

  1. Farzin, M. A.; Abdoos, H., A critical review on quantum dots: From synthesis toward applications in electrochemical biosensors for determination of disease-related biomolecules. Talanta 2021, 224, 121828.
  2. Azam, N.; Najabat Ali, M.; Javaid Khan, T., Carbon quantum dots for biomedical applications: review and analysis. Front. Mater. 2021, 8, 700403.
  3. Vitukhnovsky, A.; Zvagelsky, R.; Kolymagin, D.; Pisarenko, A.; Chubich, D., Three-Dimensional Optical Lithography and Nanoscale Optical Connectors. Bull. Russ. Acad. Sci.: Phys. 2020, 84, 760-765
  4. Arzhanov, A.; Savostianov, A.; Magaryan, K.; Karimullin, K.; Naumov, A., Photonics of semiconductor quantum dots: Applied aspects. Photonics Russ 2022, 16, 96-112.

  1. B) Correct some misprints in the text, e.g. “…oxidation effect prevails overthe quantum confinement…” – “over the” (Abstract, line 4); “32. 32…” in the list of references etc.

Response for question B

Based on the reviewer's comment, we have addressed the mistakes in the abstract and references.

Additionally, we have thoroughly reviewed the references, correcting any errors or inconsistencies that were identified. We appreciate the reviewer's diligence in pointing out these issues, and we believe that the revised manuscript now accurately reflects the intended content.

  1. C) Φ should be replaced by “diameter” or D (page 2, line 19; page 3, lines 15 and 16).

Response for question C

In response to the reviewer's suggestion, “Φ” symbol in the manuscript is revised to the “diameter” in the page 2, line 63 and page 3, line 110. We have similarly replaced the symbol with the appropriate term. We appreciate the reviewer's attention to detail, and we believe that these revisions have enhanced the clarity and accuracy of the manuscript.

  1. D) Figures 2-9 should be increased by the size and centered.

Response for question D

Thank you for your valuable suggestion. We have carefully considered your comment and have made the necessary adjustments to the sizes of Figure 2 to Figure 9 in the manuscript. We have amplified and centered the figures to improve their visibility and ensure that they are more effectively presented.
